# Statistical Regeneration Guarantees of the Wasserstein Autoencoder with Latent Space Consistency

**Anish Chakrabarty**
Statistics and Mathematics Unit
Indian Statistical Institute, Kolkata
chakrabarty.anish@gmail.com

**Swagatam Das**
Electronics and Communication Sciences Unit
Indian Statistical Institute, Kolkata
swagatam.das@isical.ac.in

## Abstract

The introduction of Variational Autoencoders (VAE) has been marked as a breakthrough in the history of representation learning models. Besides having several accolades of its own, VAE has successfully flagged off a series of inventions in the form of its immediate successors. Wasserstein Autoencoder (WAE), being an heir to that realm carries with it all of the goodness and heightened generative promises, matching even the generative adversarial networks (GANs). Needless to say, recent years have witnessed a remarkable resurgence in statistical analyses of the GANs. Similar examinations for Autoencoders, however, despite their diverse applicability and notable empirical performance, remain largely absent. To close this gap, in this paper, we investigate the statistical properties of WAE. Firstly, we provide statistical guarantees that WAE achieves the target distribution in the latent space, utilizing the Vapnik–Chervonenkis (VC) theory. The main result, consequently ensures the regeneration of the input distribution, harnessing the potential offered by Optimal Transport of measures under the Wasserstein metric. This study, in turn, hints at the class of distributions WAE can reconstruct after suffering a compression in the form of a latent law.

## 1   Introduction

Unsupervised data generation has been a time-hallowed problem in machine learning theory. Though not its strongest suit, Variational Autoencoders (VAE) [1] have been implemented for generative purposes to a great extent. Once emerged as a crucial unsupervised method for representation learning, its poor performances in this particular area somewhat depreciate its appeal. On the contrary, generative adversarial networks (GANs) [2] have proven to be superior when it comes to the quality of the generated image samples. The theoretical elegance of the adversarial training method can be duly credited for this achievement. In an attempt to adopt this merit, such 'adversarial loss' functions were incorporated into VAE objectives [3, 4]. The Wasserstein metric, also being used in GANs [5] previously, was an inevitable choice, since it brings along the beckoning of Transportation Theory. Using the same to measure the reconstruction loss in a VAE, Wasserstein Autoencoder (WAE) [6] was conceived. By approaching generative modeling from an Optimal Transport (OT) point of view, it not only achieved remarkable reconstruction standards but also strengthened the theoretical backbone.

An appropriate WAE framework consists of two components, namely an *encoder* ($E$) and a *decoder* ($D$), which are represented by neural networks in practice. Samples from an input distribution ($P_X$) are fed into the encoder, which aims at resulting in a target law ($P_Z$) in the latent space ($\mathcal{Z}$). The decoder deals with the reconstruction of the original distribution from this 'encoded' signal. Mathematically, the objective of WAEs is to minimize $\mathcal{L} = \mathcal{R}(D, E) + \Omega(E)$ over the

set of probabilistic encoders $E$. Here, $\mathcal{R}()$ ensures the reconstruction of samples put in whereas, $\Omega()$ ensures the attainment of the latent distribution. Needless to say, $\mathcal{R}()$ is represented by the Wasserstein discrepancy.

As the name suggests, our primary interest lies in investigating whether a suitably characterized WAE can *learn* a distribution, which represents its latent behaviour, without losing the *memory* of the initial signal it was fed. When it comes to the question of memorizing the input distribution in the form of reconstruction, theoretical guarantees in the context of GANs, have been provided rigorously. In the absence of a flexible discriminator, WAEs need more care and closer observation. Moreover, the concurrent task of sculpting a desired latent distribution adds nuances to its mechanism. Most of the existing literature on deep generative models, under adversarial loss, misses out on this very point. Against such a backdrop, this first of its kind study substantiates the simultaneous performances of WAE, from a theoretical viewpoint.

Our analysis starts with a special family of WAEs, namely $f$-Wasserstein Autoencoders ($f$-WAE) [7], which arises due to the specific choice of $\Omega()$ as $f$-divergences (1). We follow a non-parametric approach in defining both the networks involved and the distributions under consideration. The transformations induced by the encoder and the decoder are conceived as measurable functions between spaces. The nature of an 'ideal' encoder in our language, is a network that preserves information. Whereas, the decoder is characterized by a map that propagates the information retained, precisely. Our statistical assumptions on the same reflect these very ideas. As a result, the two concurrent objectives of $f$-WAE boil down to non-parametric estimation of laws, under vastly different circumstances.

Our major findings can be summarized in the following way,

- Given a non-negative margin of error, $f$-WAEs can learn latent space distributions with a convergence rate of $\mathcal{O}(n^{-\frac{1}{2}})$, under fairly gentle assumptions on the input and target law (1). The proof involves a realistic characterization of the transformation, induced by the encoder network.

- Under a geometry preserving representation of the decoder map, Hölder densities can be successfully reconstructed by $f$-WAEs up to a constant margin (2). Our work also produces deterministic concentration bounds on the regeneration error and hence, the rate of convergence of the empirical laws.

- A particular instance of our analysis goes on to show that the input distribution converges strongly to its latent counterpart under encoding, in the form of empirical measures. Also, $f$-WAEs can exactly reproduce samples from smooth densities in the sense of weak convergence (4).

- Inferences, identical to that in case of $f$-WAE, can be drawn for the original WAE [6], under additional assumptions on the decoder (2).

## 2 Related Work

In recent times, scrutiny of the empirical achievements of deep generative models from a theoretical point of view has gained great momentum. GANs [2], being at the helm, have received most of the attention [8, 9]. Possibly, the cornerstone of success for a GAN lies in its adversarial training and the accompanying loss functions. A plethora of GAN variants have been devised by incorporating different losses, such as Wasserstein distance (WD) [5], $f$-divergence [10], Maximum Mean Discrepancy [11] and Sobolev distances [12], to name a few. The generative aspect, in particular, has attracted a lot of research in the form of distributional estimation studies [13, 14]. Additionally, [15] has proved weak convergence of distributions under several of the aforementioned adversarial losses. In a recent study, [16] has also discussed the convergence of the empirical measures in the case of WGAN, under parametrized network assumptions. VAEs, despite being used vastly as generative tools, have not experienced such a level of independent theoretical investigation, which comprehends its latent space mechanism.

In a novel approach to unify the deep generative techniques, Adversarial AEs [3] were introduced. Subsequent work to establish resemblance [17, 4], has only strengthened that foundation. Eventually, the formulation of WAE [6] bridged the gap significantly by matching regeneration standards. The equivalence in performance is a direct reflection of the similarity of its objective to that of GANs' [7].

This 'primal-dual relationship' prepares a basis to explore the generative facets of WAE, in particular, now with the collective experience provided by GANs.

Typically, WD serves as the measure of discrepancy used to represent the reconstruction loss in WAEs. Non-parametric density estimation under this metric has seen the establishment of minimax bounds based on covering numbers, under fairly general assumptions [18, 19]. The same has also been achieved by [20], for unbounded metric spaces, assuming that some moments are finite. Singh *et al.* [21] established minimax convergence rates in the same context, under smoothness assumptions on the networks employed in the generation. Despite being influential in their own ways, none of the above aim at discussing the coinciding objectives of WAEs. In a recent empirical study, Diffusion VAE [22] was introduced, which attempts at capturing arbitrary latent manifolds using kernels of Brownian motion. However, without precise conceptual insight. As such, the theoretical questions regarding the parallel machinery of WAEs remain largely unanswered. Our analysis is a humble attempt at filling this void of statistical understanding.

# 3 Preliminaries

## 3.1 Notations

We start with denoting the input space by $\mathcal{X}$, which can be assumed to be Polish (i.e., separable and completely metrizable). We will use $\mathcal{Z}$ for the latent space. Both may be thought of as Euclidean spaces, since it is a typical characterization. The space of probability measures (distributions) defined on $\mathcal{X}$ is denoted by $\mathscr{P}(\mathcal{X})$. The set of all measurable functions from $\mathcal{X}$ to $\mathcal{Z}$ is refered to as $\mathscr{F}(\mathcal{X}, \mathcal{Z})$. For example, observe that a probabilistic encoder $E$ belongs to $\mathscr{F}(\mathcal{X}, \mathscr{P}(\mathcal{Z}))$. For any function $T : \mathcal{X} \rightarrow \mathcal{Z}$, we denote the *pushforward measure* of $\mu$ as $T_{\#}\mu \in \mathscr{P}(\mathcal{Z})$, i.e. for any measurable set $\omega \subset \mathcal{Z}$ we have $T_{\#}\mu(\omega) = \mu(\{x \in \mathcal{X} : T(x) \in \mathcal{Z}\}) = \mu(T^{-1}(\omega))$. To denote *absolute continuity* of a measure $\mu$ with respect to $\lambda$, we use the notation $\mu \ll \lambda$. The *Support* of $\mu$ is denoted by $Supp(\mu)$. To indicate the *Closure* of $\mathcal{X}$ we use $\bar{\mathcal{X}}$ and $Int(\mathcal{X})$ denotes the set of its *Interior Points*. Given $\mathcal{S} \subseteq \mathcal{X}$, the $\mathcal{S}$-*distance* between a pair of functions $f$ and $g$ is given by, $\|f - g\|_{\mathcal{S}} = \sup_{x \in \mathcal{S}}|f(x) - g(x)|$. When the sup is taken over the whole space, we call it the *uniform norm*. The *total variation distance* between probability measures $\mu$ and $\gamma \in \mathscr{P}(\mathcal{X})$ is denoted by $\|\mu - \gamma\|_{TV}$. Now we introduce some useful concepts that are made use of later.

**Definition 1** (*f*-Divergence). *Let* $P, Q \in \mathscr{P}(\mathcal{X})$ *such that* $P \ll Q$. *For a convex function* $f$ *with* $f(1) = 0$, *the f-divergence of* $P$ *from* $Q$ *is defined as*

$$D_f(P, Q) = \int_{\mathcal{X}} f\left(\frac{dP}{dQ}\right) dQ.$$

**Definition 2** (Yatracos Family [23]). *For a class of real-valued functions* $\mathcal{F}$, *defined on* $\mathcal{Z}$, *the Yatracos family associated to* $\mathcal{F}$ *is a collection of subsets of* $\mathcal{Z}$ *defined as*

$$\mathcal{Y}(\mathcal{F}) = \{x \in \mathcal{Z} : f(x) \geq g(x); \ f, g \in \mathcal{F}\}.$$

**Definition 3** (Empirical Yatracos Minimizer [24]). *The Empirical Yatracos Minimizer is a function* $\mathcal{L}^{\mathscr{P}(\mathcal{Z})} : \bigcup_{n=1}^{\infty} \mathcal{Z}^n \longrightarrow \mathscr{P}(\mathcal{Z})$ *defind as*

$$\mathcal{L}_n^{\mathscr{P}(\mathcal{Z})} = \operatorname*{argmin}_{q \in \mathscr{P}(\mathcal{Z})} \|q - \hat{p}_n\|_{\mathcal{Y}(\mathscr{P}(\mathcal{Z}))},$$

*where* $\hat{p}_n$ *is an empirical distribution based on a sequence of observations* $\{Z_i\}_{i=1}^n$ *from* $\mathcal{Z}$.

**Definition 4** ($\epsilon$-Closeness). *A pair of distributions* $f, g \in \mathscr{P}(\mathcal{X})$ *are said to be* $\epsilon$-*close to each other with respect to the metric* $c$ *if* $c(f, g) \leq \epsilon$.

**Definition 5** (Quasi-Isometric Embedding). *A map* $f : (\mathcal{Z}, d_1) \longrightarrow (\mathcal{X}, d_2)$ *is called a quasi-isometric embedding if there exists a constant* $A \geq 1$ *such that,*

$$\frac{1}{A}d_1(x, y) \leq d_2(f(x), f(y)) \leq Ad_1(x, y) \ \forall x, y \in \mathcal{Z}.$$

**Definition 6** (Covering Number [18]). *Let* $S \subseteq \mathcal{X}$. *Then, the minimum integer* $m$ *such that there exists* $m$ *closed balls* $\{B_i\}_{i=1}^m$ *of diameter* $\epsilon$ *each* $\ni S \subseteq \bigcup_{1 \leq i \leq m} B_i$, *is called the* $\epsilon$-*covering number of* $S$. *We denote it by* $\mathcal{N}_{\epsilon}(S)$.

## 3.2 Wasserstein Duality: A Brief Exposition

The equivalence relation between Wasserstein Distance and Integral Probability Metrics (IPMs) [25] is a fairly well-studied topic in optimal transport theory. We exploit this fact to serve our purpose. For a detailed elucidation on this subject, one may take refuge in [26, 27].

Let $P, Q \in \mathscr{P}(\mathcal{X})$ and $c : \mathcal{X} \times \mathcal{X} \to \mathbb{R}_{\geq 0}$ forms a metric. Then,

$$
\begin{aligned}
W_c(P, Q) &= \inf_{\gamma \in \Gamma(P,Q)} \left\{ \int_{\mathcal{X} \times \mathcal{X}} c(x, y) d\gamma(x, y) \right\} \\
&= \sup_{l \in \mathscr{L}_c^1} \left\{ \int_{\mathcal{X}} l(x) dP(x) - \int_{\mathcal{X}} l(x) dQ(x) \right\} = \text{IPM}_{\mathscr{L}_c^1}(P, Q),
\end{aligned}
$$

where $\Gamma(P, Q) = \left\{ \gamma \in \mathscr{P}(\mathcal{X} \times \mathcal{X}) : \int_{\mathcal{X}} \gamma(x, y) dy = P, \int_{\mathcal{X}} \gamma(x, y) dx = Q \right\}$ denotes the coupling between $P$ and $Q$. The class of 1-Lipschitz functions with respect to the metric $c$ is represented by $\mathscr{L}_c^1$. Later in the study however, we will use the notation $d_{\mathscr{L}_c^1}$ for Wasserstein metric, for clarity.

## 3.3 Problem Setup

A typical WAE architecture intends to solve the minimization problem,

$$
\text{WAE}_{c,\lambda}(P_X, D) = \inf_{E \in \mathscr{F}(\mathcal{X}, \mathscr{P}(\mathcal{Z}))} \left\{ \int_{\mathcal{X}} \mathbb{E}_{z \sim E(x)}[c(x, D(z))] dP_X(x) + \lambda.\Omega(E_\# P_X, P_Z) \right\}, \quad (1)
$$

where $c : \mathcal{X} \times \mathcal{X} \to \mathbb{R}_{\geq 0}$ and $\lambda > 0$ serves as a hyperparameter. Arbitrary divergence metrics between measures, belonging to $\mathscr{P}(\mathcal{Z})$, are deployed in the form of $\Omega$. In this paper however, we modify this formulation to facilitate theoretical analyses. The reshaped objective that our study is based on is given by,

$$
f\text{-WAE}_{c,\lambda}(P_X, D) = \inf_{E \in \mathscr{F}(\mathcal{X}, \mathscr{P}(\mathcal{Z}))} \left\{ W_c(P_X, (D \circ E)_\# P_X) + \lambda.D_f(E_\# P_X, P_Z) \right\}. \quad (2)
$$

This category of AEs are called $f$-Wasserstein Autoencoders [7]. The key difference between this family and the ordinary WAE arises due to the choice of the reconstruction loss. $W_c$ is the manifestation of the expected cost of transportation, over all couplings of the pair of distributions under consideration. This alteration, however, does not impact the characterization significantly. Equivalence between these two systems can be attained by imposing mild restrictions on the decoder transformation, which we elaborate on at a later stage. As such, this can be marked as a valid foundation to work on.

Observe that, the regularizer metric used in (2), to measure the discrepancy between the target latent distribution and the encoded signal is taken to be $D_f$. In this analysis, our specific choice of $f$-Divergence is Total Variation Distance (TV). Its rich theoretical appeal motivates us to do so. TV as a measure of deviation is symmetric with respect to the distributions under consideration. Moreover, it can pose both as a $f$-Divergence and an IPM [28]. The original WAE, in one of its characterizations, uses Jensen–Shannon divergence (JS) to penalize the loss instead [6]. TV, on the other hand, acts as an upper bound to JS [29], which also justifies our choice.

**Remark.** *WAEs act as a generalization to AAEs [3]. We stress on the fact that, a specific choice of $c$ as $L_2^2$ in (1), establishes an equivalence between the two [6]. Keeping that in mind, our analysis does not restrict the choice of $c$ and hence can also cater to the theory of AAEs.*

## 4 Theoretical Analysis

Before moving forward, let us rewrite the minimization problem (2) in the so-called constrained form

$$
\inf_{E \in \mathscr{F}(\mathcal{X}, \mathscr{P}(\mathcal{Z}))} \left\{ W_c(P_X, (D \circ E)_\# P_X) \right\} \text{ subject to } D_f(E_\# P_X, P_Z) \leq t, \quad (3)
$$

for some $t \geq 0$. By Lagrangian duality [30], for every value of $t$ where the constraint $D_f(E_\# P_X, P_Z) \leq t$ holds, there exists a value of $\lambda$ that produces the same solution from (2). Conversely, the solution to the problem (2), say $\hat{E}$ also solves the bound version with $t = D_f(\hat{E}_\# P_X, P_Z)$.

As such, there is an one-to-one correspondence between the constrained problem (3) and the Lagrangian one (2). This formulation gives us a direction to move ahead to establish regeneration guarantees and consistency of measures in the latent space.

Our analysis consists of two stages, going forward. Firstly we show that probabilistic encoders can successfully transform input densities into realistically characterized latent space laws, under TV distance, up to a constant margin of error. The mild assumptions imposed in the process are fairly practical. Once the constraint of the minimization problem is met in the form of latent space consistency, we provide deterministic upper bounds to the empirical reconstruction loss.

Throughout our discussion, we denote with $\mu \in \mathscr{P}(\mathcal{X})$ the input data distribution. The target distribution which characterizes the latent space is denoted by $\rho$. For simplicity, we assume that both $\mu$ and $\rho$ happen to have corresponding densities, namely $p_\mu$ and $p_\rho$ with respect to the Lebesgue measure $\lambda$, in their respective spaces. Both densities need explicit characterization, which we state in the form of certain assumptions.

**Assumption 1.** $p_\mu$, (i) based on a compact and convex support is (ii) bounded away from 0 by a constant factor $c > 0$, i.e. $\inf_x p_\mu(x) \geq c$.

**Assumption 2.** $p_\rho$, (i) also defined on a compact support, is easy-to-sample-from and (ii) infinitely smooth, i.e. $p_\rho \in C^\infty(\mathcal{Z})$.

The input density being far from nullity, passes on the same nature to the reconstructed law, even under near-ideal situations. This poses as a measure to prevent the regenerated law from being a degenerated one, at zero. We point out that, in the non-parametric approach of this work, we often use empirical distributions corresponding to $\rho$, which are based on observations from the same. In high-dimensional statistics, there are distributions from which it is practically not feasible to draw samples. For example, the drawing of uniform samples from a high-dimensional unit sphere. Although in our analysis, the need for such samples is strictly theoretical and the practical feasibility is somewhat immaterial, we describe $\rho$ this way to avoid complication. Given these assumptions, suppose the 'encoded' pushforward measure $E_\#\mu \in \mathscr{P}(\mathcal{Z})$ has density $p_{E_\#\mu}$. Then, every such $p_{E_\#\mu}$ is a potential candidate to represent $p_\rho$. However, our analysis does not assume the existence of a density. We denote $Supp(p_\rho)$ by $\mathcal{C}$, such that

**Assumption 3.** VC-dim$[\mathcal{Y}(\mathscr{P}(\mathcal{C}))]$ is finite.

This assumption too is very much plausible in its own right. For example, let $\mathcal{G}_d$ and $\mathcal{A}_d$ denote the class of $d$-dimensional Gaussian distributions and the class of $d$-dimensional axis-aligned Gaussian distributions respectively. It can be shown that, VC-dim$[\mathcal{Y}(\mathcal{G}_d)] = \mathcal{O}(d^2)$ and VC-dim$[\mathcal{Y}(\mathcal{A}_d)] = \mathcal{O}(d)$ [31]. The introduction of Yatracos class of distributions, in turn, facilitates the search for the minimum distance estimate of a measure. For a detailed exposition on the same, one may turn to [24].

Perhaps the most challenging of tasks in studying the theoretical aspects of deep neural networks lies in their basic characterization. Several depictions of networks as a collection of functions, indexed by parametric classes have produced notable results in the past [9, 13]. This paper rather explores the mathematical properties a network has to embody to function in a WAE system. The following representation of an encoder map reflects our motivation. We consider an $E \in \mathscr{F}(\mathcal{X}, \mathscr{P}(\mathcal{Z}))$ such that,

**Assumption 4.** (i) $E_\#\mu$ is supported on $\mathcal{C}$ and is also easy to draw samples from.

(ii) Given any arbitrary empirical measure $\hat{\mu}_n$ (based on $n$ samples) corresponding to $\mu$, there exists an empirical counterpart $\widehat{(E_\#\mu)}_n$ of $E_\#\mu$ which is $\epsilon$-close to $\hat{\mu}_n$ under $E$, with probability at least $1 - k\exp\{-n^r\epsilon^2\}$, where $r \geq 1$ and $k \geq 0$. In other words, $\mathbb{P}\left( \left\| E_\#\hat{\mu}_n - \widehat{(E_\#\mu)}_n \right\|_{TV} \leq \epsilon \right) \geq 1 - k\exp\{-n^r\epsilon^2\}$.

Part (ii) of Assumption (4) is provided only to give coherence to the definition of $E$. It suggests that the probability of 'the discrepancy between the two quantities involved being arbitrarily small' increases exponentially with the $r$-th power of $n$. As such, if one has a superlinear rate of increment in information in the form of samples, the portion lost in transition decays sharply. This embodies our idea of an encoder, that preserves information. It is often seen in practice that AEs, with ReLU encoders, can reconstruct images accurately, despite having the encoded observations to be almost degenerate. In such a case, the two measures under consideration can always be said to be arbitrarily

close. Part (ii) is a gentler approximation of the same idea. We now present the main result that enables one to prove consistency in estimating the latent space distribution.

**Theorem 1.** *Let, $\widehat{(E_\#\mu)}_n$ stands for an empirical measure based on $n \geq 1$ i.i.d. samples from $E_\#\mu$. Then, under assumptions [1(i), 2(i), 4(i)], for positive constants $c_1, c_2, c_3$ and $\delta \in (0, 1)$ there exists $\lambda^* \geq 0$ such that,*

$$\left\|\rho - \widehat{(E_\#\mu)}_n\right\|_{TV} - \lambda^* \leq \frac{1}{\sqrt{n}}\left\{c_1\sqrt{v} + c_2\sqrt{\ln\left(\frac{c_3}{\delta}\right)}\right\}$$

*holds with probability $\geq 1 - \delta$, where $v = $ VC-dim$[\mathcal{Y}(\mathscr{P}(\mathcal{C}))]$.*

To prove this theorem, we introduce some crucial results from the estimation theory of measures. Proofs of all the lemmas stated henceforth can be found in the supplement.

**Lemma 1.** *For $f, g \in \mathscr{P}(\mathcal{C}), \|f - g\|_{TV} = \|f - g\|_{\mathcal{Y}(\mathscr{P}(\mathcal{C}))}$.*

**Lemma 2.** *Let $\gamma \in \mathscr{P}(\mathcal{Z})$. Also, let $\hat{\gamma}_n$ denote the empirical distribution based on $n \geq 0$ i.i.d. samples drawn from $\gamma$. Then there exist positive constants $k_1, k_2$ such that*

$$\mathbb{P}\left(\|\gamma - \hat{\gamma}_n\|_{\mathcal{Y}(\mathscr{P}(\mathcal{Z}))} \leq k_1\sqrt{\frac{v}{n}} + t\right) \geq 1 - \exp\left(-k_2nt^2\right),$$

*where $v$ stands for VC-dim$[\mathcal{Y}(\mathscr{P}(\mathcal{Z}))]$.*

*Proof of Theorem 1.* Given $n$ observations from $\rho$, let $\hat{\rho}_n$ denote the corresponding empirical measure. Now, using the triangle inequality we have,

$$\left\|\rho - \widehat{(E_\#\mu)}_n\right\|_{TV} \leq \left\|\rho - E_\#\mu\right\|_{TV} + \left\|E_\#\mu - \widehat{(E_\#\mu)}_n\right\|_{TV}$$

$$\leq \|\rho - \hat{\rho}_n\|_{TV} + \left\|E_\#\mu - \widehat{(E_\#\mu)}_n\right\|_{TV} + \|\hat{\rho}_n - E_\#\mu\|_{TV}. \tag{4}$$

Define,

$$\mathcal{L}_n^* = \operatorname*{argmin}_{q \in \mathscr{P}(\mathcal{C})} \|q - \hat{\rho}_n\|_{\mathcal{Y}(\mathscr{P}(\mathcal{C}))}.$$

Observe that,

$$\|\hat{\rho}_n - E_\#\mu\|_{TV} \leq \|\hat{\rho}_n - \mathcal{L}_n^*\|_{TV} + \|\mathcal{L}_n^* - E_\#\mu\|_{TV}$$

$$= \|\hat{\rho}_n - \mathcal{L}_n^*\|_{\mathcal{Y}(\mathscr{P}(\mathcal{C}))} + \|\mathcal{L}_n^* - E_\#\mu\|_{TV} \tag{5}$$

$$\leq \|\rho - \hat{\rho}_n\|_{\mathcal{Y}(\mathscr{P}(\mathcal{C}))} + \|\mathcal{L}_n^* - E_\#\mu\|_{TV}. \tag{6}$$

Lemma (1) ensures the equality in (5). (6) is due to the definition of $\mathcal{L}_n^*$.

Now,

$$\|\mathcal{L}_n^* - E_\#\mu\|_{TV} = \inf_{\tau \in \mathcal{T}(\mathcal{L}_n^*, E_\#\mu)} \int c(x, y)d\tau(x, y) \tag{7}$$

$$= \inf_{\tau \in \mathcal{T}'(\mathcal{L}_n^*, \mu)} \int c(x, E(y))d\tau(x, y), \tag{8}$$

where $\mathcal{T}$ and $\mathcal{T}'$ are couplings of measures. The metric $c : \mathcal{Z} \times \mathcal{Z} \rightarrow \mathbb{R}_{\geq 0}$ in (7) is the trivial metric $c(x, y) = 1_{x \neq y}$ [26]. This implies, the transformation induced by the encoder should be such that, it minimizes the adversarial loss (8). In case $c$ is lower semi-continuous, there exists a $\tau \in \mathcal{T}'$ which minimizes (8), and hence an $E$ [32]. For example, $c(x, y) = 1_C(x, y)$, where $1_C(x, y)$ is the indicator on an open set $C$. In our situation however, we can always choose a sub-optimal $E$ such that (8) $\leq \lambda^*$, for a non-negative $\lambda^*$. Villani *et al.* [26] describes various ways to get hold of such a $\tau$.

Going back to (4),

$$\left\|\rho - \widehat{(E_\#\mu)}_n\right\|_{TV} - \lambda^* \leq 2\|\rho - \hat{\rho}_n\|_{\mathcal{Y}(\mathscr{P}(\mathcal{C}))} + \left\|E_\#\mu - \widehat{(E_\#\mu)}_n\right\|_{\mathcal{Y}(\mathscr{P}(\mathcal{C}))}. \tag{9}$$

Thus, (9) along with lemma (2) proves the theorem. $\qquad\square$

It is worth mentioning that in reality however, we only have a sample of size $n \in \mathbb{N}^+$ from the input distribution, say $\{X_i\}_{i=1}^n$. Based on the same, an empirical distribution $\hat{\mu}_n = \frac{1}{n}\sum_{i=1}^n \delta_{X_i}$ is what we have at hand.

**Corollary 1.** *Under the additional assumptions [3, 4(ii)],*

$$\left\|E_{\#}\hat{\mu}_n - \rho\right\|_{TV} - \lambda^* = \mathcal{O}_{\mathbb{P}}(n^{-\frac{1}{2}}).$$

**Remark.** *Despite the fact that the optimal value of (8) cannot be made arbitrarily small directly, in case we obtain $\lambda_n, n \in \mathbb{N}$, such that, $\limsup_{n \to \infty} \lambda_n = \lambda^*$; the same conclusion can be drawn. We call Corollary (1) 'consistency up to $\lambda^*$' in this light. Note that, for a given margin of error $\lambda$ in the latent space, $E$ needs to be chosen such that, $\lambda^* \leq \lambda$. On the other way round, starting with an unspecified $\lambda$, one can learn about the feasible margin of error by observing $\lambda^*$.*

Theorem (1) shows that the discrepancy between the 'encoded' and the 'target' law can be made arbitrarily small around a constant with high probability. Corollary (1), consequently establishes our idea of consistency in latent space. Proof for the same can be found in the supplement. As such, for any suitably chosen $t^*$, the constraint in problem (3) is satisfied. We now investigate whether the decoder network is capable of regenerating a distribution 'close' to $\mu$, from $E_{\#}\mu$. A probabilistic decoder, in this context, is formally defined as a function $D : \mathcal{Z} \longrightarrow \mathcal{X}$ such that $D_{\#}\rho \in \mathscr{P}(\mathcal{X})$.

The residual task at hand can be formally written as the minimization of

$$d_{\mathscr{L}_c^1}((D \circ E)_{\#}\mu, \mu) = \sup_{l \in \mathscr{L}_c^1} \left\{ \mathbb{E}_{x \sim (D \circ E)_{\#}\mu}\big[l(x)\big] - \mathbb{E}_{x \sim \mu}\big[l(x)\big] \right\}$$

over $E \in \mathscr{F}(\mathcal{X}, \mathscr{P}(\mathcal{Z}))$ which follows assumption (4). Which, however under an observed empirical measure $\hat{\mu}_n$ boils down to its sample counterpart, given by

$$d_{\mathscr{L}_c^1}((D \circ E)_{\#}\hat{\mu}_n, \mu) = \sup_{l \in \mathscr{L}_c^1} \left\{ \mathbb{E}_{x \sim (D \circ E)_{\#}\hat{\mu}_n}\big[l(x)\big] - \mathbb{E}_{x \sim \mu}\big[l(x)\big] \right\}. \tag{10}$$

We remark that the role of the class of functions $\mathscr{L}_c^1$ while minimizing (10) is to behave as 'critics'. GANs, during their training, gradually learn to be a better judge by casting this functional class through a discriminator [5]. This freedom of choice in turn gives one more flexibility to interpret its regenerative capabilities [13, 9, 14]. In the case of AEs however, the choice depends solely on the preassigned discrepancy metric. Hence, the theoretical justification of reconstruction for AEs needs additional maneuvering. Our first step in doing so is an assumption on the input distribution.

**Assumption 5.** *(i) $p_\mu \in \mathcal{H}^\alpha(\mathcal{X})$ with $\alpha > 0$, where $\mathcal{H}^\alpha(\mathcal{X})$ denotes the Hölder class based on $\mathcal{X}$; such that, the $i^{th}$ order derivative $p_\mu^{(i)}$ exists $(i \leq \lfloor\alpha\rfloor)$ and is bounded on $\bar{\mathcal{X}}$. Also,*
$D_{p_\mu}^{\lfloor\alpha\rfloor} = \sup_{x,y} \frac{\left|p_\mu^{(\lfloor\alpha\rfloor)}(x) - p_\mu^{(\lfloor\alpha\rfloor)}(y)\right|}{\|x-y\|_2^{\alpha-\lfloor\alpha\rfloor}} < \infty,$ *for $x \neq y \in Int(\mathcal{X})$.*

**Remark.** *Assumption (5) enables one to assign the Hölder norm, defined as*

$$\left\|p_\mu\right\|_{\mathcal{H}^\alpha} = \max_{i \leq \lfloor\alpha\rfloor}\left\|p_\mu^{(i)}\right\|_\infty + D_{p_\mu}^{\lfloor\alpha\rfloor}.$$

*Moreover, $\left\|p_\mu\right\|_{\mathcal{H}^\alpha}$ turns out to be bounded.*

Smoothness assumptions on input data distribution, in the form of Hölder or Sobolev densities are quite familiar in the theory of density estimation for generative models [13, 14, 21]. Observe that, earlier results still hold under this additional assumption. The smoothness of both the input and latent densities (2(ii)) together ensures the existence of certain regenerative maps, which are crucial to the forthcoming discussion. Lemma (3) formalizes the same.

**Lemma 3** ([33])**.** *Under assumptions $[1, 2, 5]$, there exists a map $T \in \mathcal{H}^{\alpha+1}$ from $\mathcal{Z}$ to $\mathcal{X}$ such that, $T_{\#}\rho = \mu$.*

In our discussion, similar to the encoder transformation, we do not impose specific structural restrictions on the decoder network. However, the mathematical assumptions that are made, are inspired and supported by real architectures.

**Assumption 6.** *The quasi-isometric embedding $D : (\mathcal{Z}, \|\ \|_{TV}) \longrightarrow (\mathcal{X}, c)$ induced by the decoder network is $\epsilon$-close to $T$ in uniform norm, where $T$ is as defined in lemma (3). Also, $(D \circ E)_{\#}\mu$ shares the same support with $\mu$.*

Classes of functions, learned by generator networks in GANs, are often taken to be smooth [13, 21] to achieve minimax convergence rate. Our assumption of quasi-isometric embedding is gentler in principle. We mention that an alternative weaker assumption of Lipschitz continuity would have sufficed in our analysis. Assumption (6) on the other hand, takes one step forward to comment on the broader geometry of the two spaces under consideration. Moreover, precise specification of parameters of a ReLU network can always result in a map, which is capable of approximating the optimal Monge solution [34, 35], which in our case turns out to be smooth. Note that, the metric on $\mathcal{Z}$ can be equivalently taken as $\| \ \|_1$.

The following lemma provides an oracle inequality by fragmenting the empirical loss as in (10), into disjoint parts.

**Lemma 4.** *Given an encoder transformation $E^*$,*

$$d_{\mathscr{L}_c^1}((D \circ E^*)_{\#}\hat{\mu}_n, \mu) \leq \mathcal{E}_1 + \mathcal{E}_2 + 2\mathcal{E}_3,$$

*where $\mathcal{E}_1 = d_{\mathscr{L}_c^1}((D \circ E^*)_{\#}\hat{\mu}_n, D_{\#}\rho)$ is the 'Encoder Approximation Error';*
*$\mathcal{E}_2 = d_{\mathscr{L}_c^1}(D_{\#}\rho, T_{\#}\rho)$ stands for the 'Decoder Approximation Error' and*
*$\mathcal{E}_3 = d_{\mathscr{L}_c^1}(\hat{\mu}_n, \mu)$, the 'Statistical Estimation Error'.*

We remark that successful regeneration by the $f$-WAE can be ensured by showing: these individual errors are arbitrarily small or they approach zero in any probabilistic sense. To demonstrate the same in the case of $\mathcal{E}_3$, we introduce some concepts from [36, 18].

**Definition 7** (1-Upper Wasserstein Dimension)**.** *For $\gamma \in \mathscr{P}(\mathcal{X})$, the $(\epsilon, \tau)$-covering number is given by*

$$\mathcal{N}_\epsilon(\gamma, \tau) = \inf_{S \subseteq \mathcal{X}} \{\mathcal{N}_\epsilon(S) : \gamma(S) \geq 1 - \tau\}.$$

*Consequently, we define the $(\epsilon, \tau)$-dimension as $\delta_\epsilon(\gamma, \tau) = \frac{\mathcal{N}_\epsilon(\gamma, \tau)}{-\log(\epsilon)}$. Under this setup, the Upper Wasserstein Dimension for $p = 1$ is*

$$\delta_1^*(\gamma) = \inf\{s \in (2, \infty) : \limsup_{\epsilon \to 0} \delta_\epsilon(\gamma, \epsilon^{\frac{s}{s-2}}) \leq s\}.$$

**Lemma 5** ([18])**.** *Let $diam_c(Supp(\mu)) = B$. Then, for $n \geq 0$ and $s > \delta_1^*(\mu)$,*

$$\mathbb{P}\left(d_{\mathscr{L}_c^1}(\hat{\mu}_n, \mu) \leq \mathcal{O}(n^{-\frac{1}{s}}) + t\right) \geq 1 - \exp\left\{-\frac{2nt^2}{B^2}\right\}.$$

Now, we propose the main theorem which provides a guarantee regarding regeneration.

**Theorem 2.** *Given a margin of error $\lambda^*$ in latent space, let $\hat{\mu}_n$ denote an empirical input measure, consistent up to $\lambda^*$, under encoder map $E^*$. Also, let $diam_c(Supp(\mu)) = B$. Then, there exists a constant $\zeta^* \geq 0$ and $\delta \in (0, 1)$ such that*

$$d_{\mathscr{L}_c^1}((D \circ E^*)_{\#}\hat{\mu}_n, \mu) - \zeta^* \leq \mathcal{O}(n^{-\frac{1}{s}}) + \mathcal{O}(n^{-\frac{1}{2}})$$

*with probability at least $1 - \delta$, where $s > \delta_1^*(\mu)$.*

*Proof of Theorem 2.* Observe that, the error committed by the encoder

$$\mathcal{E}_1 = d_{\mathscr{L}_c^1}((D \circ E^*)_{\#}\hat{\mu}_n, D_{\#}\rho) = \inf_{\upsilon \in \Upsilon(E_{\#}^*\hat{\mu}_n, \rho)} \int c(D(x), D(y))d\upsilon(x, y) \qquad (11)$$

$$\leq A \inf_{\upsilon \in \Upsilon(E_{\#}^*\hat{\mu}_n, \rho)} \int \|x - y\|_{TV} \, d\upsilon(x, y)$$

$$\leq \lambda^* A B + \mathcal{O}(n^{-\frac{1}{2}}), \qquad (12)$$

where (12) holds with probability $\geq 1 - \delta$ (using Theorem (1)). $\Upsilon$ in (11) denotes the associated coupling and $A \geq 1$ is the constant related to the embedding $D$. Denote, $\zeta^* = \lambda^* A B$.

Similarly,

$$\mathcal{E}_2 = d_{\mathscr{L}_c^1}(D_{\#}\rho, T_{\#}\rho) = \sup_{l \in \mathscr{L}_c^1} \left\{\mathbb{E}_{x \sim \rho}\big[l(D(x))\big] - \mathbb{E}_{x \sim \rho}\big[l(T(x))\big]\right\}$$

$$\leq \mathbb{E}_{x \sim \rho}\big[\|D - T\|_\infty\big] \leq \epsilon^*, \qquad (13)$$

which is a consequence of the fact that, $\|D - T\|_\infty \leq \epsilon$ (Assumption (6)). Here, $\epsilon^*$ is an arbitrarily small quantity corresponding to $\epsilon$.

Also, lemma (5) suggests that for $s > \delta_1^*(\mu)$,

$$\mathcal{E}_3 = d_{\mathscr{L}_c^1}(\hat{\mu}_n, \mu) \leq \mathcal{O}(n^{-\frac{1}{s}}) + \frac{1}{\sqrt{n}} B \sqrt{\frac{1}{2} \ln\left(\frac{1}{\delta}\right)} \tag{14}$$

with probability at least $1 - \delta$.

As such, (12), (13) and (14) together prove the theorem. □

**Remark.** *Let us weigh up on a special case, which carries notable significance. Suppose, one obtains $\lambda_n$ as an upper bound to (8), corresponding to $\mathcal{L}_n^*$, $n \in \mathbb{N}^+$, such that $\lambda_n = o(1)$. We do not prove the existence of such a sequence, which might be taken up as a future work. However, under the assumption of existence, corollary (1) implies that,*

$$\sup_{\mathcal{A} \in \mathcal{F}} \left\| E_\# \hat{\mu}_n(\mathcal{A}) - \rho(\mathcal{A}) \right\| \longrightarrow 0,$$

*where $\mathcal{F}$ is the Borel $\sigma$-algebra on $\mathcal{Z}$. As such, $E_\# \hat{\mu}_n$ converges strongly to $\rho$. This theoretically ensures the achievement of a target latent law.*

*Similarly, such an $\{\lambda_n\}$ will have a remarkable effect on the regeneration process. In that case, Theorem (2) will imply,*

$$d_{\mathscr{L}_c^1}((D \circ E^*)_\# \hat{\mu}_n, \mu) \longrightarrow 0,$$

*i.e. $(D \circ E^*)_\# \hat{\mu}_n$ converges weakly in $\mathscr{P}(\mathcal{X})$ to $\mu$ [26]. This result serves as an exact reconstruction guarantee.*

Now, let us look back at the original WAE objective in (1). Denote by

$$d_c^*(P_X, (E, D)) = \int_{\mathcal{X}} \mathbb{E}_{z \sim E(x)}[c(x, D(z))] dP_X(x)$$

the reconstruction loss. In general, this acts as an upper bound to the $f$-WAE counterpart [7]. However, under additional restrictions on D, they become equivalent [7]. In such a case, if $\Omega$ in (1) is taken to be $D_f$, the two objectives in their entirety also follow the same trait. In our study, the measure of discrepancy between distributions in latent space was throughout taken to be TV. JS divergence, being upper bounded by TV, is obliged to follow the results established exactly. As such, the only task remaining is the transferral of regeneration guarantees to WAEs.

**Corollary 2.** *There exists a specific margin of error $\kappa \geq 0$ in the latent space such that, Theorem (2) holds exactly for $d_c^*(\hat{\mu}_n, (E^*, D))$, where D is also invertible.*

*Proof of Corollary (2).* Husain *et al.* [7] had suggested the existence of a $\kappa \geq 0$, such that for all margin or error $\lambda > \kappa$, (1) and (2) are exactly equal. The proof however, demands the decoder transform to be invertible. In such a case, for an encoder $E^*$ and $\Omega = D_f$ imply that

$$d_c^*(\hat{\mu}_n, (E^*, D)) = \int_{\mathcal{X}} \mathbb{E}_{z \sim E^*(x)}[c(x, D(z))] d\hat{\mu}_n(x) \leq d_{\mathscr{L}_c^1}((D \circ E^*)_\# \hat{\mu}_n, \mu) + d_{\mathscr{L}_c^1}(\hat{\mu}_n, \mu).$$

Hence, using lemma (2) and Theorem (2) we say that

$$d_c^*(\hat{\mu}_n, (E^*, D)) - \zeta^* \leq \mathcal{O}(n^{-\frac{1}{s}}) + \mathcal{O}(n^{-\frac{1}{2}})$$

holds with probability at least $1 - \delta$, for $s > \delta_1^*(\mu)$. □

We emphasize the fact that the argument of weak convergence, however, as given in (4), does not apply directly to this case. The Wasserstein distance metrizes $\mathscr{P}(\mathcal{X})$, i.e. weak convergence holds if and only if $d_{\mathscr{L}_c^1} \longrightarrow 0$. On the other hand, $d_c^*$ does not satisfy the axioms of a distance. As such, WD produces stronger inferences, even though under carefully taken assumptions they tend to be equal.

## 5 Further Theoretical Implications

An important aspect of the VAE's operations lies in its involvement in 'disentanglement'. While the concept lacks a robust definition, it can arguably be described as the process of achieving a factorial representation; one with 'expressive' and perhaps independent coordinates, which characterize the features. A somewhat gentle depiction of the same might be a distribution with a diagonal dispersion matrix. The non-parametric portrayal of latent distributions in this analysis does not comment on the covariance structure of the underlying random variables. However, this seemingly unrestrained characterization in turn may allow for such specific distributions also, as far as they follow the assumptions of our framework. For example, we mention that if $\mathscr{P}(\mathcal{C})$ is taken to be the class of finite-dimensional axis-aligned Gaussians, our results hold true.

A practitioner might also feel divided between the two architectures that appear in [6]. Although we focus mainly on the one deploying JS divergence, there is no evidence of theoretical superiority that MMD enjoys, to the best of our knowledge. Uniform convergence bounds on empirical MMD between distributions, as established by [37] in a different context, however, also produce a convergence rate of $\mathcal{O}(n^{-\frac{1}{2}})$. As such, this section can be regarded as a note of encouragement for closer observations on these very topics.

## 6 Conclusion

Our analysis, besides serving as the first of its kind theoretical promises regarding the simultaneous tasks a WAE has to perform, also points toward untraveled avenues. An immediate extension might look into broader classes of discrepancy measures in latent space. The latent space and hence the distribution that characterizes it might also be in focus. An even elaborate characterization can deliver it further justice. Our study includes a commentary on the geometry preserving nature of deep generative networks from a fairly new perspective. Being a novel representation, it creates new opportunities for inventive scrutiny. We also restrict our analysis to a non-parametric regime, which may encourage similar endeavours under a parametrized setup. Such an approach could be the key to revealing the effect of network parameters such as depth, width, activation functions etc. on regeneration and consistency in the latent space. This, in turn, may improve the theoretical results by allowing training plans on board as well.

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

## Funding Transparency Statement

The authors did not receive any third-party funding or third-party support during the last 36 months prior to this submission toward pursuing this work. The authors have no financial relationships with entities that could potentially be perceived to influence the submitted work.

