# Supplement to: "Statistical Regeneration Guarantees of the Wasserstein Autoencoder with Latent Space Consistency"

**Anish Chakrabarty**
Statistics and Mathematics Unit
Indian Statistical Institute, Kolkata

**Swagatam Das**
Electronics and Communication Sciences Unit
Indian Statistical Institute, Kolkata

## A   Appendix

*Proof of lemma (1).*  Here, $\mathscr{P}(\mathcal{C})$ denotes the set of probability measures defined on the common support $\mathcal{C}$. This is a slight abuse of the notation, since $\mathcal{C}$ is not the underlying space, but a subset of the $\sigma$-algebra defined on it. Consequently,

$$\mathcal{Y}(\mathscr{P}(\mathcal{C})) = \big\{\omega \in \mathcal{C} : f_1(\omega) \geq f_2(\omega); \; f_1, f_2 \in \mathscr{P}(\mathcal{C})\big\}.$$

Let, $f, g \in \mathscr{P}(\mathcal{C})$. Observe that,

$$\sup_{\omega \in \mathcal{C}}\big|f(\omega) - g(\omega)\big| = \|f - g\|_{TV} \geq \|f - g\|_{\mathcal{Y}(\mathscr{P}(\mathcal{C}))},$$

due to the definition of TV.

Define, $A = \{\omega \in \mathcal{C} : f(\omega) \geq g(\omega)\} \in \mathcal{Y}(\mathscr{P}(\mathcal{C}))$. Now,

$$\|f - g\|_{TV} = \frac{1}{2}\|f - g\|_1 = \big|f(A) - g(A)\big| \leq \|f - g\|_{\mathcal{Y}(\mathscr{P}(\mathcal{C}))}.$$

∎

*Proof of lemma (2).*  Since we only deal with measures supported on $\mathcal{C}$, our proof revolves around $\mathscr{P}(\mathcal{C})$. A similar argument will hold for all the measures, based on the $\sigma$-algebra corresponding to $\mathcal{Z}$.

Let, $\gamma \in \mathscr{P}(\mathcal{C})$. Also, let $\{X_i\}_{i=1}^n$ denote an i.i.d. sample from $\gamma$. Define, $\hat{\gamma}_n(S) = \frac{1}{n}\sum_{i=1}^n \delta_{X_i}(S)$, for $S \in \mathcal{C}$.

Using Dudley's chaining argument coupled with symmetrization, it can be shown that (Corollary 7.18 [1]) there exists an universal constant $L$ such that,

$$\mathbb{E}\Big[\sup_{S \in \mathcal{Y}(\mathscr{P}(\mathcal{C}))}\big|\hat{\gamma}_n(S) - \gamma(S)\big|\Big] \leq L\sqrt{\frac{\text{VC-dim}[\mathcal{Y}(\mathscr{P}(\mathcal{C}))]}{n}}.$$

This constant $L$ depends on the diameter of $\mathcal{C}$ with respect to the $\|\;\|_2$ norm. Now, by McDiarmid's inequality

$$\mathbb{P}\Big(\sup_{S \in \mathcal{Y}(\mathscr{P}(\mathcal{C}))}\big|\hat{\gamma}_n(S) - \gamma(S)\big| - \mathbb{E}\big[\sup_{S \in \mathcal{Y}(\mathscr{P}(\mathcal{C}))}\big|\hat{\gamma}_n(S) - \gamma(S)\big|\big] \geq \eta\Big) \leq \exp\left(-cn\eta^2\right),$$

where $c$ is a positive constant. As such,

$$\mathbb{P}\Big(\|\hat{\gamma}_n - \gamma\|_{\mathcal{Y}(\mathscr{P}(\mathcal{C}))} \geq L\sqrt{\frac{v}{n}} + \eta\Big) \leq \exp\left(-cn\eta^2\right)$$

$$\Longleftrightarrow \mathbb{P}\Big(\|\hat{\gamma}_n - \gamma\|_{\mathcal{Y}(\mathscr{P}(\mathcal{C}))} \leq L\sqrt{\frac{v}{n}} + \frac{1}{\sqrt{n}}\sqrt{\frac{1}{c}\ln\left(\frac{1}{\delta}\right)}\Big) \geq 1 - \delta,$$

where $v = \text{VC-dim}[\mathcal{Y}(\mathscr{P}(\mathcal{C}))]$ and $\delta \in (0, 1)$. Judicious choices of $k_1$ and $k_2$ proves the lemma.  ∎

35th Conference on Neural Information Processing Systems (NeurIPS 2021).

*Proof of lemma (4).* Since, Wasserstein distance is a metric on $\mathscr{P}(\mathcal{X})$, using triangle inequality we get

$$
\begin{aligned}
d_{\mathscr{L}_c^1}((D \circ E^*)_\# \hat{\mu}_n, \mu) &\leq d_{\mathscr{L}_c^1}((D \circ E^*)_\# \hat{\mu}_n, \hat{\mu}_n) + d_{\mathscr{L}_c^1}(\hat{\mu}_n, \mu) \\
&\leq d_{\mathscr{L}_c^1}((D \circ E^*)_\# \hat{\mu}_n, D_\# \rho) + d_{\mathscr{L}_c^1}(D_\# \rho, \hat{\mu}_n) + \mathcal{E}_3 \\
&\leq d_{\mathscr{L}_c^1}(D_\# \rho, T_\# \rho) + d_{\mathscr{L}_c^1}(T_\# \rho, \hat{\mu}_n) + \mathcal{E}_1 + \mathcal{E}_3 \\
&= \mathcal{E}_1 + \mathcal{E}_2 + 2\mathcal{E}_3.
\end{aligned}
$$

Here, $T$ is as suggested in lemma (3). ∎

*Proof of lemma (5).* Theorem 1 of [2] ensures that, for $s > \delta_1^*(\mu)$

$$
\mathbb{E}\big[d_{\mathscr{L}_c^1}(\hat{\mu}_n, \mu)\big] = \mathcal{O}(n^{-\frac{1}{s}}).
$$

Denote, $W(\omega) = d_{\mathscr{L}_c^1}(\hat{\mu}_n, \mu)$, where $\omega \in \mathcal{X}^n$. Now, for $x_1, x_2, ..., x_n, x_n' \in \mathcal{X}$

$$
\left| W(x_1, x_2, ..., x_n) - W(x_1, x_2, ..., x_n') \right| \leq \frac{1}{n} c(x_n, x_n') \leq \frac{B}{n}.
$$

As such, $d_{\mathscr{L}_c^1}(\ )$ satisfies the bounded difference inequality. Thus, using the McDiarmid's inequality we get

$$
\mathbb{P}\Big(d_{\mathscr{L}_c^1}(\hat{\mu}_n, \mu) - \mathbb{E}\big[d_{\mathscr{L}_c^1}(\hat{\mu}_n, \mu)\big] \geq t\Big) \leq \exp\Big\{ - \frac{2nt^2}{B^2}\Big\},
$$

$t > 0$ i.e., $\big\{d_{\mathscr{L}_c^1}(\hat{\mu}_n, \mu) \leq \mathcal{O}(n^{-\frac{1}{s}}) + t\big\}$ holds with probability at least $1 - \exp\big( - \frac{2nt^2}{B^2}\big)$. ∎

*Proof of Corollary (1).* Observe that,

$$
\mathbb{P}\Big( \big\| E_\# \hat{\mu}_n - \rho \big\|_{TV} - \lambda^* - c_1 \sqrt{\frac{v}{n}} \geq \epsilon \Big)
$$

$$
\leq \mathbb{P}\Big( \Big\| E_\# \hat{\mu}_n - \widehat{(E_\# \mu)}_n \Big\|_{TV} + \Big\| \widehat{(E_\# \mu)}_n - \rho \Big\|_{TV} - \lambda^* - c_1 \sqrt{\frac{v}{n}} \geq \epsilon \Big)
$$

$$
\leq \mathbb{P}\Big( \Big\| E_\# \hat{\mu}_n - \widehat{(E_\# \mu)}_n \Big\|_{TV} \geq \frac{\epsilon}{2} \Big) + \mathbb{P}\Big( \Big\| \widehat{(E_\# \mu)}_n - \rho \Big\|_{TV} - \lambda^* - c_1 \sqrt{\frac{v}{n}} \geq \frac{\epsilon}{2} \Big)
$$

$$
\leq k \exp\Big\{ - \frac{n^r \epsilon^2}{4}\Big\} + c_3 \exp\Big\{ - \frac{n c' \epsilon^2}{4}\Big\}, \tag{1}
$$

where $c' = \frac{1}{c_2^2}$ and $v = \text{VC-dim}[\mathcal{Y}(\mathscr{P}(\mathcal{C}))]$, which is taken to be finite. Theorem (1) and Assumption (4(ii)) together result in (1). Hence, for $r \geq 1$, $c^* = \min\{\frac{1}{4}, \frac{c'}{4}\}$ and $k^* = 2\max\{k, c_3\}$,

$$
\mathbb{P}\Big( \big\| E_\# \hat{\mu}_n - \rho \big\|_{TV} - \lambda^* \geq c_1 \sqrt{\frac{v}{n}} + \epsilon \Big) \leq k^* \exp\Big\{ - n c^* \epsilon^2 \Big\}.
$$

i.e., with probability at least $1 - \delta$,

$$
\big\| E_\# \hat{\mu}_n - \rho \big\|_{TV} - \lambda^* \leq \mathcal{O}(n^{-\frac{1}{2}}) + \frac{1}{\sqrt{n}} \sqrt{\frac{1}{c^*} \ln\big(\frac{k^*}{\delta}\big)}.
$$

∎

**Remark** (Regarding Proof of lemma (3))**.** *The objective at hand is to find a $T : \mathcal{Z} \longrightarrow \mathcal{X}$ such that,*

$$
T \in \underset{T: T_\# \rho = \mu}{\arg\min} \int c(x, T(x)) d\rho(x).
$$

*Assumption (1) and (5) ensure that the density corresponding to $\mu$ is smooth in the sense of Hölder and is based on a convex $\mathcal{X}$. $p_\rho$ has also been taken to be smooth (2). When $\mathcal{X}, \mathcal{Z} \subseteq \mathbb{R}^d$, a quadratic cost $c$ implies that such a solution $T$ exists (Brenier Potential) and moreover, satisfies the Monge-Ampère equation (Eq. 12.4 in [3]). In this premise, the regularity results on $T$, provided by Caffarelli et al.[4] exactly proves Lemma (3).*