# OpenReview forum: "Statistical Regeneration Guarantees of the Wasserstein Autoencoder with Latent Space Consistency"
_NeurIPS.cc/2021/Conference — NeurIPS 2021 Spotlight_

### Official Review · Reviewer_TEbm · 2021-07-13

**Rating:** 6
**Confidence:** 4

**Summary:**

This paper provides a statistical guarantee of f-WAEs on learning the latent distribution by VC theory.

**Limitations And Societal Impact:**

It's also better to discuss if there is any technical contribution. For the proof of Theorem 1, it looks naive and just uses the basic concentration. In assumption 4, "$E_{\#}\mu$ is easy to draw sample from". This is not a rigorous statement.

**Main Review:**

This paper is well-written with accurate and proper definition on related notations and probability measures which is appreciated.

The author studied statistical properties of f-WAE on learning latent space and reconstruction. This is the first kind of such result from my knowledge.

My main concern is about the implication of those theoretical results.

--- The current theoretical result is a bit rough and hard for the practitioner to gain some insights why WAE is better than vanilla VAE and AEs.

--- The main theoretical results in Theorems 1 and 2 are independent of training and specific model architecture. For example, whether the VC-dimension will be related to the depth and width of the networks? In Theorem 1, how large lambda^* it is? Is that an absolute constant or can be arbitrarily large? Do you have any model assumption on the networks? For me, assumption 4 is a bit strong.  I suggest the author could discuss under what cases it can be satisfied.

**Time Spent Reviewing:**

5

---

> ### Author Response · Authors · 2021-08-09
> **Response**
>
> We thank the reviewer for the insightful and constructive comments. We are much delighted to learn that the reviewer has appreciated some aspects of the paper and found it "well-written with accurate and proper definition on related notations and probability measures".
>
> Please find our response to your **concerns regarding the implication of the theoretical results** below.
>
> --- VAEs, though empirically shown to be capable, lack statistical justification similar to ours. WAEs, on the other hand, were introduced with the promise of heightened generative performance. They offer enhanced flexibility over VAEs in terms of the choice of discrepancy measure between the latent and encoded distributions. Our work, taking advantage of this very fact, proves latent space consistency using the TV metric. Moreover, the employment of Wasserstein metric as the reconstruction objective leads to sharp uniform concentration bounds on the empirical loss, eventually establishing regeneration guarantees. It also allows for a transportation theoretic viewpoint of the problem. Also, a specific choice of the cost $c$ as $L_{2}^2$ makes WAEs equivalent to the famous Adversarial Autoencoders (AAEs) [1] that attempt to elegantly blend the autoencoder architecture with the adversarial loss concept embraced by GANs. Our analysis does not restrict the choice of $c$, and hence can also cater to the theory of AAEs.
>
> [1] Alireza Makhzani, Jonathon Shlens, Navdeep Jaitly, Ian Goodfellow and Brendan Frey. Adversarial Autoencoders. *arxiv*:1511.05644, ICLR 2016 Workshop.
>
> --- In this work, we intend to develop a theoretical framework that identifies encoder and decoder networks as transformations between spaces. This in turn enables one to introduce transportation theoretic operators, e.g. pushforward measures. The characterizations of such transformations, however, are inspired by realistic network architectures. Moreover, our approach is non-parametric in nature; and specifying network architectures or particular training plans will bring along model parameters that need to be addressed. The introduction of training may also give rise to the question of domain adaptation and additional errors due to differences between training and testing data. We feel such questions are quite different from our objective and need separate scrutiny to be served justice. To answer the specific question of VC dimension, however, we would like to point out that it is an intrinsic property of the class of distributions defined on the latent space and hence independent of network depth and width.
>
> For a given margin of error in the latent space $\lambda$, the constant $\lambda^*$ should be $\leq \lambda$. As the proof of Theorem 1 suggests, construction of a coupling $\tau$ accordingly may lead to such a $\lambda^*$. On the other way round, starting with an unspecified $\lambda$, one can learn about the feasible margin of error by observing $\lambda^*$.
>
> Assumption (4) can be viewed as a criterion which ensures that the empirical measure corresponding to the encoded input distribution can be replaced by an encoded empirical distribution corresponding to the input law, without committing huge errors. It is often seen in practice that AEs, with ReLU encoders, are able to reconstruct images accurately, despite having the encoded observations to be almost degenerate. This fact in particular poses a challenge while formulating clustering techniques in latent space after using AEs for dimensionality reduction. The data points being arbitrarily close to each other always tend to hail from the same cluster, in other words from the same law. In such a case, the two measures under consideration can always be said to be arbitrarily close. Assumption (4) is a gentler approximation of the same idea.
>
> Below are our response to the comments under **Limitations And Societal Impact** head.
>
> --- This analysis is the first of its kind and we believe is capable of offering a framework to approach the theory of AEs from a new and improved perspective. In lemma 4, we introduce a novel oracle inequality for the reconstruction error, which deconstructs the same into disjoint components. Moreover, a statistical assurance of such a vastly used method like WAE goes beyond empirical evidence to attest its worth, which our work delivers. Note that some connections explored in this paper with the native concepts of high-dimensional statistics and density estimation could be of potential interest to the machine learning community for further theoretical analysis of deep generative models in the future as well. We also stress the fact that the uniform concentration bounds shown in the paper offer an uncomplicated yet elegant way to prove the results. They also accommodate for the measures of richness of the underlying spaces efficiently, serving our purpose.
>
> In our non-parametric approach, we often use empirical distributions corresponding to $E_{\\#}\mu$ which are based on observations from the same. In high-dimensional statistics, there are distributions from which it is practically not feasible to draw samples, e.g. drawing uniform samples from a high dimensional unit sphere. Although in our analysis, the need for such samples is strictly theoretical and the practical feasibility is somewhat immaterial. We describe $E_{\\#}\mu$ this way for easier understanding.
>
> We pledge to try our best to address the suggestions by the reviewer in our final version. We hope, taking into account the other reviews and our responses, you will consider revising the score for this paper.

---

> > ### Comment · Area_Chair_FQSf · 2021-08-12
> > **Reviewer please reply and if warranted update review**
> >
> > Your AC

---

> > ### Comment · Reviewer_TEbm · 2021-08-15
> > **response is clear**
> >
> > The authors clearly address my questions. I am raising my score to 6.

---

### Official Review · Reviewer_sBV7 · 2021-07-13

**Rating:** 7
**Confidence:** 4

**Summary:**

The submission aims to provide rigorous theoretical results for the Wasserstein Autoencoder (WAE) - A model that has achieved great empirical success yet remains severely understudied. In particular, their main contribution in showing learning theoretic bounds such as complexity and convergence guarantees. As a feature of the Autoencoder set-up, they introduce the notion of statistical regeneration and go out to prove this statement for their learning setting.

**Limitations And Societal Impact:**

Yes they have.

**Main Review:**

The main strength of this paper is the problem they tackle which is an extremely topical issue with little theoretical studies, especially for the autoencoder settings. The contribution of the paper is original and makes important and clear references to related work when appropriate. The significance of the paper is also clear by virtue of the importance of the problem.

In particular, the authors establish a connection to the VC-dimension of the encoder distribution classes which is an interesting manifestion of complexity in the autoencoder setting. This opens up fruitful future work in the direction of studying autoencoders and generative models are large from a learning theoretic perspective. The work also establishes the notion of statistical regeneration, which although somewhat not interpretable, can be an important step for understanding the generalization performance of autoencoder models.

The clarity of the paper can definitely be improved as I found it quite challenging to follow notation and had to refer to external references to better understand some definitions. For example, in Assumption 3, I could not properly understand what is this space \mathcal{Y}( \mathscr{P}(\mathcal{C})) formally? I understand it refers to the choice of models however I would like to better understand what this function class is precisely. However, I understand that the area of generative modelling and autoencoders are quite notation dense, given the differing ways they are defined so this is not so much an issue for a paper with such results.

Some questions I had for the authors:

(1) One of the main interesting questions about WAE is how should one pick the divergence between distributions in the latent space. This paper comments on this however I was wondering if there is anything extremely actionable that you can comment? I am aware that the most common choices are MMD due to computational convenience and perhaps this has some stronger effect on the convergence?

(2) There is a common discussion in the autoencoder literature between the choice of lambda (the regularization weighting) and 'disentanglement'. While disentanglement is not properly defined, I wonder if the results of your work can establish this formally from the perspective of statistical regeneration since it seems to be intimately linked with the latent space. Do you have any comments on this?



**Time Spent Reviewing:**

13

---

> ### Author Response · Authors · 2021-08-09
> **Response**
>
> We thank the reviewer for the inspiring and pertinent comments. Please find our response to your queries below.
>
> As the reviewer has pointed out, the analysis in its entirety demands a large number of concepts to be introduced with respective definitions. In the final version, we will indeed provide more expository notes and/or suitable references to better describe the technical terms and notations thereby improving the clarity.
>
> The concept of Yatracos class is native to the theory of density estimation and is used to facilitate the search for the minimum distance estimate. Please note that the family $\mathcal{Y}( \mathscr{P}(\mathcal{C}))$ denotes the collection of Scheffe sets, of distributions based on $\mathcal{C}$ (please see the reference [1] below). The intrinsic ordering while defining this class in turn helps to choose the best estimate corresponding to an empirical measure. One advantage of using the Scheffe sets is that one can immediately utilize the VC dimension of the corresponding Yatracos class to bound the empirical error.
>
> [1] Luc Devroye and Gábor Lugosi. Combinatorial Methods in Density Estimation. Springer Series in Statistics. Springer-Verlag, New York, 2001.
>
>
>
> **(1)** Arguably, it is the abundance of choice in divergence metrics that makes WAEs suited to diverse applications, also enriching it theoretically. In practice, Jensen-Shannon divergence (JS), Kullback-Leibler (KL) divergence and MMD, all have shown remarkable performances. However, it might be unwise to dismiss the possibility that other *f*-divergences also perform well.
>
> To the best of our knowledge, there is no evidence of theoretical superiority that MMD enjoys. Uniform convergence bounds on empirical MMD between distributions, as established by [2] in a different context, however, produces similar results to that shown in our work. They also report a convergence rate of $\mathcal{O}(n^{-\frac{1}{2}})$, which comes in accordance with that of our analysis.
>
>
> [2] Arthur Gretton, Karsten M. Borgwardt, Malte J. Rasch, Bernhard Schölkopf, and Alexander Smola. A kernel two-sample test. JMLR 2012.
>
>
>
> **(2)** While the definition of disentanglement is not clear, to the best of our understanding, it can arguably be described as the process of achieving a factorial representation; one with 'expressive' and perhaps independent coordinates, which characterize the features. A somewhat gentle representation of the same might be a distribution with an axis-aligned covariance structure (diagonal dispersion matrix). In our work, the latent distribution is not parametrized and hence we do not comment on the covariance matrices of underlying random variables. This seemingly unrestrained characterization in turn may allow for such specific distributions also, as far as they follow the assumptions of our framework. As an example, in the paper, we mention that if the latent distribution family is taken to be the class of $d$-dimensional axis-aligned Gaussians, our results hold true.

---

> > ### Comment · Reviewer_sBV7 · 2021-08-30
> > **Thank you for the response**
> >
> > Thank you for the response, which answers my questions to greater detail than I expected. I continue to feel that this paper should be accepted at NeurIPS.

---

### Official Review · Reviewer_wdFS · 2021-07-17

**Rating:** 7
**Confidence:** 3

**Summary:**

This paper analyzes the statistical properties of WAE, providing theoretical guarantees for both sample reconstruction and latent distribution tasks. To do so, the authors use the family of so-called f-Wasserstein autoencoders due to the specific choice of discrepancy selected for measuring the target and latent distribution. Here total variation distance is chosen. After setting up necessary preliminaries, and assumptions the authors prove the first of two main theorems, consistency in estimating the latent space distribution. Under suitable conditions, the second theorem provides a guarantee regarding the regeneration. Since they analysis was conducted using TV, which upper bounds the Jenson-Shannon divergence, the final corollary transfers the regeneration guarantees to WAEs. Fairly gentle assumptions and interesting theoretical results are discussed in detail.

**Limitations And Societal Impact:**

Yes.

**Main Review:**

Originality:
This paper offers novel analysis of the simultaneous tasks of WAE, and reveled new statistical guarantees for latent space consistency and regeneration quality. To my knowledge such an analysis has not been done. The related work seems appropriately cited.

Quality:
The analysis presented here doesn’t appear to have technical issues. The work is self-contained, and the authors appear to fairly describe where any limitations may lie.

Clarity:
The submission is clearly written and well organized though some polishing could be done. Line 82-84 reads awkwardly, and I’m not sure it’s fair to say VAEs haven’t had much theoretical work done. Nevertheless, the logic from lemmas to theorems is nicely built up, with discussions where needed. I enjoyed reading it.

Significance:
Given the pace at which the AI community advances, theoretical work is often playing catch up. Results like the ones presented here are a delight, and I’d imagine others may find them interesting as well.  While it’s difficult to determine whether it advances state-of-the-art (no empirical results), it principally offers statistical guarantees of WAE using VC theory. I believe the approach is of significance and that the NeurIPS community would benefit from the contributions presented in this paper.

Typo:
Equation 10, the last parenthesis under the square root should be on the outside.


**Time Spent Reviewing:**

8

---

> ### Author Response · Authors · 2021-08-09
> **Updated Response**
>
> We thank the reviewer for the positive and inspiring comments. We are delighted to learn that the reviewer has "enjoyed reading it" and provided encouraging comments like "Results like the ones presented here are a delight".
>
> We regret the fact that the sentence in lines 82-84 lacks fluency. We intend to point out the absence of promising literature, delivering statistical explanation of the generative capabilities of VAE, which simultaneously addresses its latent space mechanism.
>
> We also appreciate the reviewer's meticulous reading of our manuscript. We pledge to undertake all the suggested rectifications in our final version.
>
> Dear Reviewer, given the mostly positive comments from your side and taking into account the other reviews and our responses, we fervently hope you will consider revising the score for this paper.

---

> > ### Comment · Reviewer_wdFS · 2021-08-21
> > **Follow up to response**
> >
> > I’ve read the other reviews as well as the responses and believe that concerns have been appropriately addressed. I will increase my score to 7.

---

### Decision · Program_Chairs · 2021-09-27

**Decision:**

Accept (Spotlight)

**Comment:**

All reviewers liked the paper and support acceptance. So it is a clear accept.

The reviewers provided good feedback on shortcomings in the presentation. The authors are strongly encouraged to take these comments into account for the final version of the paper.